# Combined analyses of within-host SARS-CoV-2 viral kinetics and information on past exposures to the virus in a human cohort identifies intrinsic differences of Omicron and Delta variants

**Timothy W. Russell**[1⊙]*, **Hermaleigh Townsley**[2,3⊙], **Sam Abbott**[1⊙], **Joel Hellewell**[4‡], **Edward J. Carr**[2,5‡], **Lloyd A. C. Chapman**[1,6], **Rachael Pung**[1], **Billy J. Quilty**[1], **David Hodgson**[1], **Ashley S. Fowler**[2], **Lorin Adams**[2], **Chris Bailey**[2,5], **Harriet V. Mears**[2], **Ruth Harvey**[2], **Bobbi Clayton**[2], **Nicola O'Reilly**[2], **Yenting Ngai**[2,5], **Jerome Nicod**[2], **Steve Gamblin**[2], **Bryan Williams**[3,5], **Sonia Gandhi**[2,5], **Charles Swanton**[2,5], **Rupert Beale**[2,5,7], **David L. V. Bauer**[2,7], **Emma C. Wall**[2,3,5‡], **Adam J. Kucharski**[1‡]

**1** Centre for Mathematical Modelling of Infectious Diseases, London School of Hygiene & Tropical Medicine, London, United Kingdom, **2** The Francis Crick Institute, London, United Kingdom, **3** National Institute for Health Research (NIHR) University College London Hospitals (UCLH) Biomedical Research Centre and NIHR UCLH Clinical Research Facility, London, United Kingdom, **4** European Molecular Biology Laboratory-European Bioinformatics Institute, Cambridge, United Kingdom, **5** University College London, London, United Kingdom, **6** Lancaster University, Bailrigg, Lancaster, United Kingdom, **7** Genotype-to-Phenotype UK National Virology Consortium (G2P-UK), London, United Kingdom

⊙ These authors contributed equally to this work.
‡ JH and EJC also contributed equally to this work. ECW and AJK also contributed equally to this work.
* timothy.russell@lshtm.ac.uk

**Data Availability Statement:** The dataset containing the Ct values at the three gene targets

## Abstract

The emergence of successive Severe Acute Respiratory Syndrome Coronavirus 2 (SARS-CoV-2) variants of concern (VOCs) during 2020 to 2022, each exhibiting increased epidemic growth relative to earlier circulating variants, has created a need to understand the drivers of such growth. However, both pathogen biology and changing host characteristics—such as varying levels of immunity—can combine to influence replication and transmission of SARS-CoV-2 within and between hosts. Disentangling the role of variant and host in individual-level viral shedding of VOCs is essential to inform Coronavirus Disease 2019 (COVID-19) planning and response and interpret past epidemic trends. Using data from a prospective observational cohort study of healthy adult volunteers undergoing weekly occupational health PCR screening, we developed a Bayesian hierarchical model to reconstruct individual-level viral kinetics and estimate how different factors shaped viral dynamics, measured by PCR cycle threshold (Ct) values over time. Jointly accounting for both interindividual variation in Ct values and complex host characteristics—such as vaccination status, exposure history, and age—we found that age and number of prior exposures had a strong influence on peak viral replication. Older individuals and those who had at least 5 prior antigen exposures to vaccination and/or infection typically had much lower levels of shedding. Moreover, we found evidence of a correlation between the speed of early shedding and duration of

considered for the 157 infection episodes, symptom onset times and covariate data, along with the code to re-run the analysis can be found at this publicly available repository: https://github.com/thimotei/legacy_ct_modelling. We have archived the stable release version of this repository on Zenodo with the following DOI: https://zenodo.org/doi/10.5281/zenodo.10223417. Furthermore, the data required to reproduce all of the figures, tables and supplementary figures can be found at this Zenodo repository DOI: https://zenodo.org/doi/10.5281/zenodo.10223385.

**Funding:** TWR, LACC and AJK were supported by funding from the Wellcome Trust (20650/Z/17/Z). AJK was also supported by the National Institutes of Health (1R01AI141534-01A1). JH was supported by a fellowship from the EMBL Interdisciplinary Postdocs — Exploring Life in context (EPIOD-LinC) programme. LACC was also supported by the National Institute for Health Research NIHR (200908). SA was funded by the Wellcome Trust (grant: 210758/Z/18/Z). RP acknowledges funding from the Singapore Ministry of Health. DH was supported by the National Institute for Health Research (NIHR; 1R01AI141534-01A1: DH). This research was partly funded by the National Institute for Health Research (NIHR) using UK aid from the UK Government to support global health research. The views expressed in this publication are those of the author(s) and not necessarily those of the NIHR or the UK Department of Health and Social Care (16/136/46: BJQ; 16/137/109: BJQ). Bill & Melinda Gates Foundation (OPP1139859: BJQ). This research was funded in whole, or in part, by the Wellcome Trust [FC011104, FC011233, FC001030, FC001159, FC001827, FC001078, FC001099, FC001169]. This work was supported by the National Institute for Health Research University College London Hospitals Department of Health NIHR Biomedical Research Centre (BRC), as well as by the UK Research and Innovation and the UK Medical Research Council (MR/W005611/1), and by the Francis Crick Institute which receives its core funding from Cancer Research UK (FC011104, FC011233, FC001030, FC001159, FC001827, FC001078, FC001099, FC001169, CC2230), the UK Medical Research Council (FC011104, FC011233, FC001030, FC001159, FC001827, FC001078, FC001099, FC001169), and the Wellcome Trust (FC011104, FC011233, FC001030, FC001159, FC001827, FC001078, FC001099, FC001169, CC2230). C.S. is a Royal Society Napier Research Professor (RSRP\R\210001). His work is supported by the Francis Crick Institute that receives its core funding from Cancer Research UK (CC2041), the UK Medical

incubation period when comparing different VOCs and age groups. Our findings illustrate the value of linking information on participant characteristics, symptom profile and infecting variant with prospective PCR sampling, and the importance of accounting for increasingly complex population exposure landscapes when analysing the viral kinetics of VOCs.

**Trial Registration:** The Legacy study is a prospective observational cohort study of healthy adult volunteers undergoing weekly occupational health PCR screening for SARS-CoV-2 at University College London Hospitals or at the Francis Crick Institute (NCT04750356) (22,23). The Legacy study was approved by London Camden and Kings Cross Health Research Authority Research and Ethics committee (IRAS number 286469). The Legacy study was approved by London Camden and Kings Cross Health Research Authority Research and Ethics committee (IRAS number 286469) and is sponsored by University College London Hospitals. Written consent was given by all participants.

## Background

Successive Severe Acute Respiratory Syndrome Coronavirus 2 (SARS-CoV-2) variants of concern (VOCs) have exhibited stepwise increases in relative epidemic growth rates [1–3]. The selective advantages of successive VOCs are the result of complex epistatic relationships arising from combinations of mutations occurring in the viral genome. Selective advantage is frequently associated with mutations in the region encoding the spike protein. To increase the growth rate of a VOC within a community, these mutations must confer improved evasion of antibody-mediated immunity, increased intrinsic transmissibility, or a combination of both. Understanding how interrelated factors including pathogen biology and host characteristics—such as preexisting immunity—impact on replication and transmission of SARS-CoV-2 is essential to inform ongoing Coronavirus Disease 2019 (COVID-19) planning and response.

Over the course of a SARS-CoV-2 infection, the viral load in the nasopharynx increases after infection until reaching a peak and then declining. The dynamics of the viral load over time are known as viral kinetics. Viral kinetics are quantified by performing reverse transcription PCR (RT-PCR) on nasal swabs, this returns a cycle threshold (Ct) value that measures the concentration of viral RNA in the nasopharynx [4]. Thus, the cycle thresholds inversely correlate with higher RNA viral concentrations. Such values measure viral RNA in the nasopharynx, which correlate well with live virus as measured by plaque-forming unit (PFU)/ml in the early stages of infection [4]. Viral kinetics will vary between individuals due to host factors such as age, sex, or prior immunity or due to changes in the pathogen related to the emergence of new VOCs [4–14].

Viral kinetics may change between variants and epidemic waves [11,14], necessitating updating of recommendations as new variants become dominant. However, as population immunity accumulates via infection and vaccination, it will become harder to generalise descriptive insights from one population to another. Studies that can identify factors influencing viral kinetics and likely infectiousness are therefore crucial for developing up-to-date public guidance and understanding risk of transmission for patients.

Changes in viral kinetics have implications for transmission since higher viral loads are frequently associated with increased host transmissibility. There are several reasons higher transmissibility could be observed after the emergence of a new VOC, including higher immune escape; faster initial replication; higher peak viral load; longer period of viral shedding; or a

Research Council (CC2041), and the Wellcome Trust (CC2041). For the purpose of Open Access, the author has applied a CC BY public copyright licence to any Author Accepted Manuscript version arising from this submission. C.S. is funded by Cancer Research UK (TRACERx (C11496/A17786), PEACE (C416/A21999) and CRUK Cancer Immunotherapy Catalyst Network); Cancer Research UK Lung Cancer Centre of Excellence (C11496/A30025); the Rosetrees Trust, Butterfield and Stoneygate Trusts; NovoNordisk Foundation (ID16584); Royal Society Professorship Enhancement Award (RP/EA/180007 & RF\ERE \231118)); National Institute for Health Research (NIHR) University College London Hospitals Biomedical Research Centre; the Cancer Research UK-University College London Centre; Experimental Cancer Medicine Centre; the Breast Cancer Research Foundation (US) (BCRF-22-157); Cancer Research UK Early Detection an Diagnosis Primer Award (Grant EDDPMA-Nov21/100034); and The Mark Foundation for Cancer Research Aspire Award (Grant 21-029-ASP). This work was supported by a Stand Up To Cancer-LUNGevity-American Lung Association Lung Cancer Interception Dream Team Translational Research Grant (Grant Number: SU2C-AACR-DT23-17 to S. M. Dubinett and A.E. Spira). Stand Up To Cancer is a division of the Entertainment Industry Foundation. Research grants are administered by the American Association for Cancer Research, the Scientific Partner of SU2C. C.S. is in receipt of an ERC Advanced Grant (PROTEUS) from the European Research Council under the European Union's Horizon 2020 research and innovation programme (grant agreement no. 835297). The funders had no role in study design, data collection and analysis, decision to publish, or preparation of the manuscript.

**Competing interests:** C.S. acknowledges grants from AstraZeneca, Boehringer-Ingelheim, Bristol Myers Squibb, Pfizer, Roche-Ventana, Invitae (previously Archer Dx Inc - collaboration in minimal residual disease sequencing technologies), Ono Pharmaceutical, and Personalis. He is Chief Investigator for the AZ MeRmaiD 1 and 2 clinical trials and is the Steering Committee Chair. He is also Co-Chief Investigator of the NHS Galleri trial funded by GRAIL and a paid member of GRAIL's Scientific Advisory Board. He receives consultant fees from Achilles Therapeutics (also SAB member), Bicycle Therapeutics (also a SAB member), Genentech, Medicxi, China Innovation Centre of Roche (CICoR) formerly Roche Innovation Centre – Shanghai, Metabomed (until July 2022), Relay Therapeutics SAB member, Saga Diagnostics SAB member and the Sarah

different symptom profile, and hence different host behaviour. As of 2023, individuals within populations have a range of life course exposure histories due to their past infections and vaccinations. These complex life course exposure histories make it challenging to determine whether changes in observed viral kinetics are due to host factors such as prior immunity or changes in the pathogen itself. Studies that can separate such factors influencing viral kinetics, and hence likely infectiousness, are therefore crucial for developing up-to-date public guidance and understanding risk of transmission for patients.

Previous studies have sought to relate within-host viral kinetics with other measurable properties such as disease severity, mortality, VOC, and the probability of onward transmission [4–6,8–10,15]. However, they have often had one of 2 key limitations. Either they have used serial sampling following an initial diagnostic swab, conducted only after symptom onset, thus missing the early rates of viral replication [13]. Or, they used single sampling with Ct values that correspond to a single time point per individual, meaning that variation in the viral load over the course of an infection is not distinguishable from the variation in viral load between individuals [16,17]. As a result, any conclusions about differences in Ct value between individuals may be an artefact of factors such as diagnostic protocols or underlying epidemic dynamics.

Repeatedly testing many individuals throughout their infection, including in the early presymptomatic period, makes it possible to separate individual Ct dynamics over time from the between-individual variation in viral load [11,18]. In cohort studies where individuals have been repeatedly tested to infer the viral dynamics of their infection, this between-individual variation has previously been compared to single characteristics, including the infecting VOC [5,11,12], age [5,7], and vaccination status [5,12]. However, there can be considerable confounding between different characteristics. For example, each sequential VOC has spread against a different background of overall population vaccine coverage, including numbers of doses received, confounding any comparisons based only on VOC. Failure to adjust for such confounding factors risks attributing differences in viral load to different VOCs that are in fact associated with factors that influence viral replication, such as the presence of vaccine-induced immune effectors such as neutralising antibody levels. Accounting for both interindividual variation in Ct values and complex host characteristics—such as vaccination status, exposure history, and age—is therefore crucial for obtaining reliable estimates of the relationship between VOCs and viral kinetics [19].

To address this complexity, we developed a Bayesian hierarchical modelling framework that estimates the unobserved Ct trajectories for each individual, using data from a subset of a prospective cohort undergoing weekly occupational health PCR screening for SARS-CoV-2. As well as accounting for different covariates jointly, this modelling approach pools data across individuals to identify the main drivers of variation in Ct dynamics and estimate the accompanying uncertainty. Using this model, we examined the relationship between Ct dynamics and different VOCs, including timing and peak viral loads, and quantified the impact of key individual-level characteristics on these dynamics.

## Methods

### Clinical cohort composition

The Legacy study is a prospective observational cohort study of healthy adult volunteers undergoing weekly occupational health PCR screening for SARS-CoV-2 at University College London Hospitals or at the Francis Crick Institute (NCT04750356) [20,21]. The Legacy study was approved by London Camden and Kings Cross Health Research Authority Research and Ethics committee (IRAS number 286469) and is sponsored by University College London Hospitals. Written consent was given by all participants. Participants provided baseline

Cannon Research Institute. C.S has received honoraria from Amgen, AstraZeneca, Bristol Myers Squibb, GlaxoSmithKline, Illumina, MSD, Novartis, Pfizer, and Roche-Ventana. C.S. has previously held stock options in Apogen Biotechnologies and GRAIL, and currently has stock options in Epic Bioscience, Bicycle Therapeutics, and has stock options and is co-founder of Achilles Therapeutics. Patents: C.S declares a patent application (PCT/US2017/028013) for methods to lung cancer); targeting neoantigens (PCT/EP2016/059401); identifying patent response to immune checkpoint blockade (PCT/EP2016/071471); methods for lung cancer detection (US20190106751A1); identifying patients who respond to cancer treatment (PCT/GB2018/051912); determining HLA LOH (PCT/GB2018/052004); predicting survival rates of patients with cancer (PCT/GB2020/050221), methods for systems and tumour monitoring (PCT/EP2022/077987). C.S. is an inventor on a European patent application (PCT/GB2017/053289) relating to assay technology to detect tumour recurrence. This patent has been licensed to a commercial entity and under their terms of employment C.S is due a revenue share of any revenue generated from such license(s).

**Abbreviations:** COVID-19, Coronavirus Disease 2019; CrI, credible interval; Ct, cycle threshold; LOD, limit of detection; LOO, leave-one-out; PFU, plaque-forming unit; RT-PCR, reverse transcription PCR; SARS-CoV-2, Severe Acute Respiratory Syndrome Coronavirus 2; VOC, variant of concern.

demographics including age, sex, and medical comorbidities, with data collection on subsequent vaccine doses and COVID-19 infections. Participants reporting either a positive test result for SARS-CoV-2 or onset of symptoms consistent with COVID-19 were sent same-day swabs via courier up to day 10 post positive test or symptom onset. They received further swabs every 2 to 3 days until day 10 (**Fig 1A**). All participants had a face-to-face postinfection study visit before day 30 postinfection, at which point an additional swab was collected, and a symptom diary was completed [22]. We linked diagnostic swab data with earlier swab data from the occupational screening pipeline at the Francis Crick Institute, enabling estimation of the presymptomatic period between the last reported negative screening swab and onset of SARS-CoV-2 infection.

We included all infection episodes reported by Legacy study participants between 3 June 2021 and 25 April 2022, which had at least 2 positive RT-PCR results. The total size of the cohort up to the maximum date considered was 680 participants, of which 152 had 2 positive results and, as such, were included in this analysis. Five individuals had 2 recorded reinfections, resulting in a total of 157 infection episodes with at least 2 positive PCR test results per episode. For each infection episode, whole-genome sequencing or a combination of date of infection and viral genotype was used to assign the VOC that caused the infection.

## SARS CoV-2 RT-PCR and viral sequencing

Viral RNA was extracted from nasopharyngeal swabs and analysed by RT-PCR using methods previously described in Aitken and colleagues' article [23]. ORF1ab, N, and S gene Ct values were used for this analysis, with an in-model adjustment for gene type. We found strong correlation between the Ct values at the different gene targets (**Fig A in S1 Text**). Therefore, including all Ct values available increased statistical power for each individual. Viral RNA from positive swabs was prepared for whole-genome sequencing using the ARTIC method (https://www.protocols.io/view/ncov-2019-sequencing-protocol-v3-locost-bh42j8ye) and sequenced on the ONT GridION platform to >30k reads/sample. The data were demultiplexed and processed using the viralrecon pipeline (https://github.com/nf-core/viralrecon) [23].

## Estimating the incubation period of SARS-CoV-2 variants

In total, 136/157 (86.6%) of infection episodes had symptoms reported, reported by their date of symptom onset (**Table 1**). Participants were a subset of the wider post-vaccine infection cohort within the Crick, where serial sampling data were available [22]. We used the difference between the estimated time of exposure and the time of symptom onset for each individual to fit 2 parameters of an assumed log-normal form of the incubation period for SARS-CoV-2 (**Fig 1B and 1C**). We specified semi-informative priors for both parameters using existing estimates of the incubation period [24].

## Estimating viral kinetics by variant and host characteristics

We constructed a Bayesian model that jointly infers individual-level Ct trajectories, population (covariate)-level trajectory parameters, parameters of a log-normally distributed incubation period, and multiplicative effect size parameters following a linear regression. We regressed against several covariates for both the population-level Ct trajectory and incubation period parameters. Covariates used are described in detail in the next section. The model allows for the addition (or removal) of covariates through a formula interface.

For the individual-level Ct trajectories, we extended and modified an existing model of individual-level Ct dynamics over the course of entire infections of any respiratory disease measurable by RT-PCR test and applied this to our cohort of participants infected with

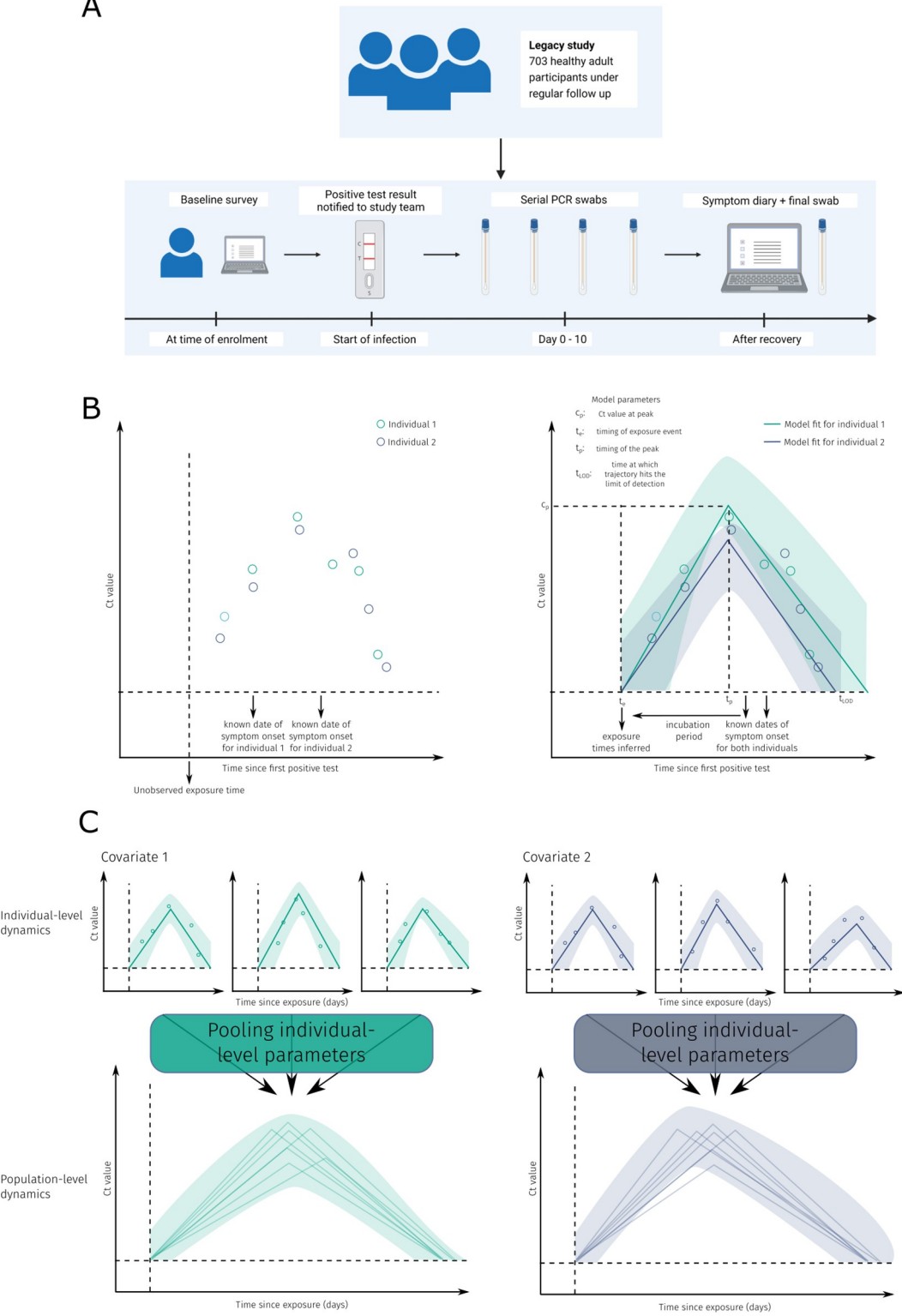

**Fig 1. Schematic of the study design and modelling procedure. (A)** Schematic of the cohort and data collection procedure used for the subset of the Legacy study. **(B)** Typical Ct value data and model fits for 2 representative individuals with different covariates. **(C)** Schematic of the individual-level and covariate-level model fits. The model fits each participant's viral kinetics and pools the estimates in a statistically robust manner. *All three of these images were drawn by the authors.*

**Table 1. Number of participants in each covariate category used in our study.** Covariate categories used in the baseline model include VOC status, symptom status, total number of exposures at time of infection, and age group. The other covariates are investigated in one of the various alternative models considered.

| Demographic details of participants[1] | | | |
|---|---|---|---|
| Characteristic | Delta, $N = 34^1$, $n = 34^2$ | Omicron BA.1, $N = 63^1$, $n = 65^2$ | Omicron BA.2, $N = 60^1$, $n = 62^2$ |
| Female[1] | 18 (53%) | 43 (68%) | 39 (65%) |
| Male[1] | 16 (47%) | 20 (32%) | 21 (35%) |
| Median age (years) [IQR] | 35 [22–48] | 40 [21–59] | 34 [17–51] |
| Number of doses at time of infection[1,3] | | | |
| 2* | 2 (3.7%) | 0 (0%) | 0 (0%) |
| 2 | 32 (59.3%) | 16 (25%) | 1 (1.7%) |
| 3* | 18 (33.3%) | 5 (8.0%) | 0 (0%) |
| 3 | 2 (3.7%) | 47 (75%) | 59 (98.3%) |
| Dose 1[1] | | | |
| AZD1222 | 16 (47.1%) | 22 (34.9%) | 17 (28.3%) |
| BNT162b2 | 18 (52.9%) | 39 (61.9%) | 38 (63.3%) |
| mRNA1272 | 0 (0%) | 2 (3.2%) | 4 (6.7%) |
| others | 0 (0%) | 0 (0%) | 1 (1.7%) |
| Dose 2[1] | | | |
| AZD1222 | 16 (47.1%) | 22 (34.9%) | 16 (26.7%) |
| BNT162b2 | 18 (52.9%) | 39 (61.9%) | 39 (65.0%) |
| mRNA1272 | 0 (0%) | 2 (3.2%) | 4 (6.7%) |
| others | 0 (0%) | 0 (0%) | 1 (1.7%) |
| Dose 3[1] | | | |
| BNT162b2 | 10 (29.4%) | 47 (74.6%) | 58 (96.6%) |
| mRNA1272 | 2 (5.9%) | 16 (25.3%) | 2 (3.4%) |
| N/A | 22 (64.7%) | 0 (11.1%) | 0 (0%) |
| Total number of exposures at time of included infection episode[1] | | | |
| 3 | 20 (58.9%) | 11 (17.4%) | 1 (1.7%) |
| 4 | 11 (32.4%) | 41 (65%) | 43 (71.7%) |
| 5+ | 3 (8.9%) | 11 (17.4%) | 16 (26.7%) |
| Joined study before infection episode?[1] | | | |
| No | 13 (38.2%) | 33 (52.3%) | 22 (36.7%) |
| Yes | 21 (61.7%) | 30 (47.6%) | 38 (63.3%) |
| Unknown | | | |
| Median days since dose prior to infection [IQR] | 121 [83.5–184] | 75 [34.0–108] | 93 [69.0–113] |
| Self-reported symptom severity[1] | | | |
| grade I | 12 (35.3%) | 28 (46.7%) | 24 (40%) |
| grade II | 11 (32.4%) | 22 (36.7%) | 33 (55%) |
| grade III | 0 (0%) | 4 (6.7%) | 0 (0%) |
| asymptomatic | 9 (26.5%) | 9 (15%) | 3 (5%) |
| Unknown | 2 (5.9%) | 0 (0%) | 0 (0%) |
| Self-reported duration of symptoms [IQR] | 10 [7–14] days | 10 [8–14] days | 11 [8–15] days |
| Virus sequenced | 32 (94%) | 56 (89%) | 32 (53%) |

[1]Number of individuals.

[2]n, infection episodes.

[3]Infection commenced within 0—14 days of dose.

(%); median [25%–75%].

SARS-CoV-2 [11,12]. The individual-level dynamics are modelled with a piecewise linear curve, on a logarithmic scale, with a single breakpoint, where the breakpoint represents the peak of the trajectory (**Fig 1B**). The logarithmic scale corresponds to assuming an exponential increase in viral load from the point of exposure until the peak, then an exponential decrease at a different rate to the point of clearance.

Each individual trajectory was parameterised using 4 parameters with the following epidemiological interpretations, each corresponding to events that were not directly observed: the timing of initial exposure event, timing of the peak (lowest) Ct value, Ct value at the peak, and the timing the trajectory hits the limit of detection (LOD, highest Ct value) (**Fig 1B and 1C**), which for this assay was Ct = 40. The reported LOD is a censored value, meaning in theory, a more sensitive machine could detect SARS-CoV-2 at higher Ct values. As such, we also estimate a true (censored) LOD for each individual, both at the start and end of each infection episode. We report differences in the peak Ct value, the timing of the peak, and the timing trajectories hit the censored LOD.

The viral kinetic model generated the expected Ct value at a given time since exposure for each individual, which was compared to the measured Ct values using a normally distributed observation process. The variance parameter of this term was shared across all individuals and represents the measurement error from the RT-PCR sampling and testing process. Given that we fit to data at 3 gene targets (ORF1ab, S, and N-gene targets), we include a term in the Bayesian linear regression adjusting for differences in gene target (**Fig A in S1 Text**).

Given the complex nature of the exposure history of each individual (**Fig 2A**), we adjusted for the time since the last exposure for each individual. We did so by calculating the time since the last infection or vaccination (whichever was sooner) for each individual as well as estimating effect sizes for the total numbers of exposures for each individual. This adjustment was performed in the Bayesian multiple linear regression component of the model, and, as such, it did not include explicit antibody dynamics.

### Covariate categories and regression reference choices

The categories of covariates used were VOC status (Delta, Omicron BA.1, or Omicron BA.2), symptom status (symptomatic or asymptomatic), age group (20 to 34, 35 to 49, and 50+), total number of antigenic exposures (3, 4, or 5+; calculated as the sum of infections and vaccine doses), and time (days) since the last exposure (see Table 1 for full description of the number of individuals in each category).

In the linear regression framework, values for VOC status, symptom status, age group, and numbers of exposures needed to be chosen for the reference case against which the other covariates were regressed. Symptomatic, Omicron BA.1 individuals, aged between 35 and 49 years with 4 exposures were chosen as the baseline case for both interpretability and statistical power, as this combination produced the largest reference group of individuals.

After fitting the model, we used the covariate-level posterior distributions to simulate 10,000 Ct trajectories for each covariate included in the analysis and summarised the trajectories by calculating the median, 50%, and 95% credible intervals for each covariate. Details and comparisons of alternative models—using either different model structures or differences in the regression formula—can be found in the Supporting information (**Figs J-M in S1 Text**).

## Results

### Differences in peak Ct value by variant and host characteristics

There was considerable variation in both individual characteristics and Ct dynamics across the cohorts (**Fig 2B**). Applying our Bayesian model to these data, we estimated that the Ct

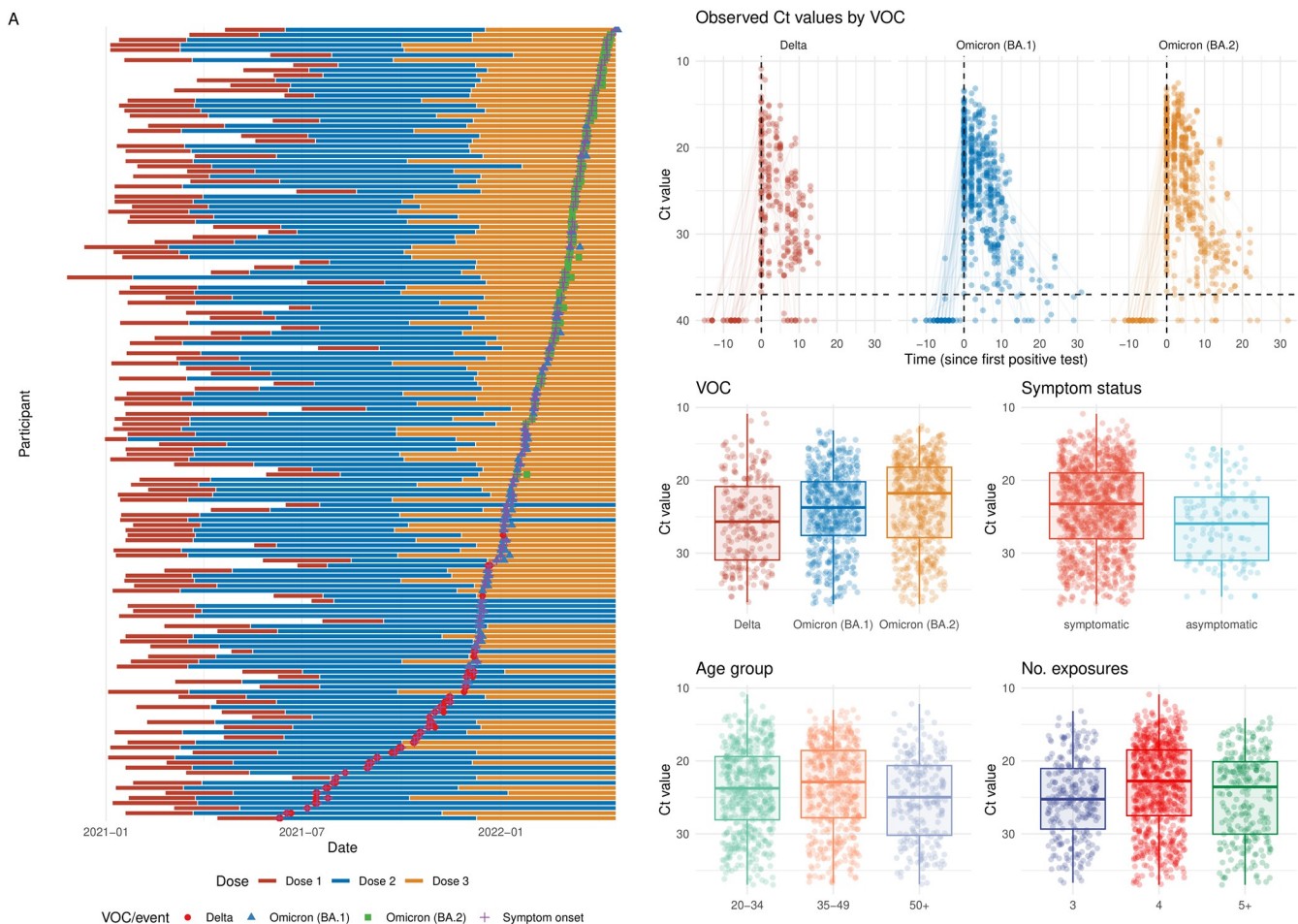

**Fig 2. Summary of longitudinal cohort dataset analysed.** (A) Ct value distribution for all participants, stratified by the interaction between the number of vaccines and the infecting VOC, the interaction between the number of infections and the infecting VOC, and, finally, by the 4 covariate categories used throughout the study: VOC, symptom status, number of exposures, and age. (B) The individual-level Ct data, including the timing of each test, the Ct value of the result, given by the colour and the number of vaccinations, infections, and the total exposures for each participant on the left of their individual timeline. Data underlying the graphs in this figure can be found in https://zenodo.org/doi/10.5281/zenodo.10223385.

trajectories for participants in the reference regression group (Omicron BA.1, symptomatic, 4 exposures, and aged 35 to 49 years) peaked at 15.9 (95% credible interval [CrI]: 14.8 to 16.9) (**Fig 3A** and **S3 Table**). In comparison, we estimated that the trajectories of the Delta- and BA.2-infected subsets have peak Ct values of 14.8 (95% CrI: 13.5 to 16.2) and 14.9 (95% CrI: 13.9 to 16.0), respectively (**Fig 3A** and **S3 Table**), resulting in differences of 0.93 (95% CrI: −0.41 to 2.40) and 0.89 (95% CrI: −0.17 to 2.0) Ct values from baseline individuals (respectively). We estimated that asymptomatic individuals have a peak Ct value of 16.6 (95% CrI: 15.1 to 18.2)—a peak Ct value 0.75 (95% CrI: −0.91 to 2.3) higher than baseline individuals (**Fig 3B** and **S4 Table**)—corresponding to a lower peak viral load. We find that individuals with 3 exposures in total have a peak Ct value of 17.0 (95% CrI: 15.3 to 18.8). These individuals and the baseline individuals have peak Ct values 1.2 (95% CrI: −0.81 to 3.2) and 2.3 (95% CrI: −1 to 3.7) lower (respectively) than individuals with 5+ exposures (who have a peak Ct value of 18.2 (95% CrI: 16.9 to 19.6)), corresponding to substantially higher viral load for both 3- and 4-exposure individuals (**Fig 3C** and **S5 Table**). Estimates for the 20 to 34 age group are almost identical to the estimates for baseline individuals. However, 50+ year olds have peak Ct value

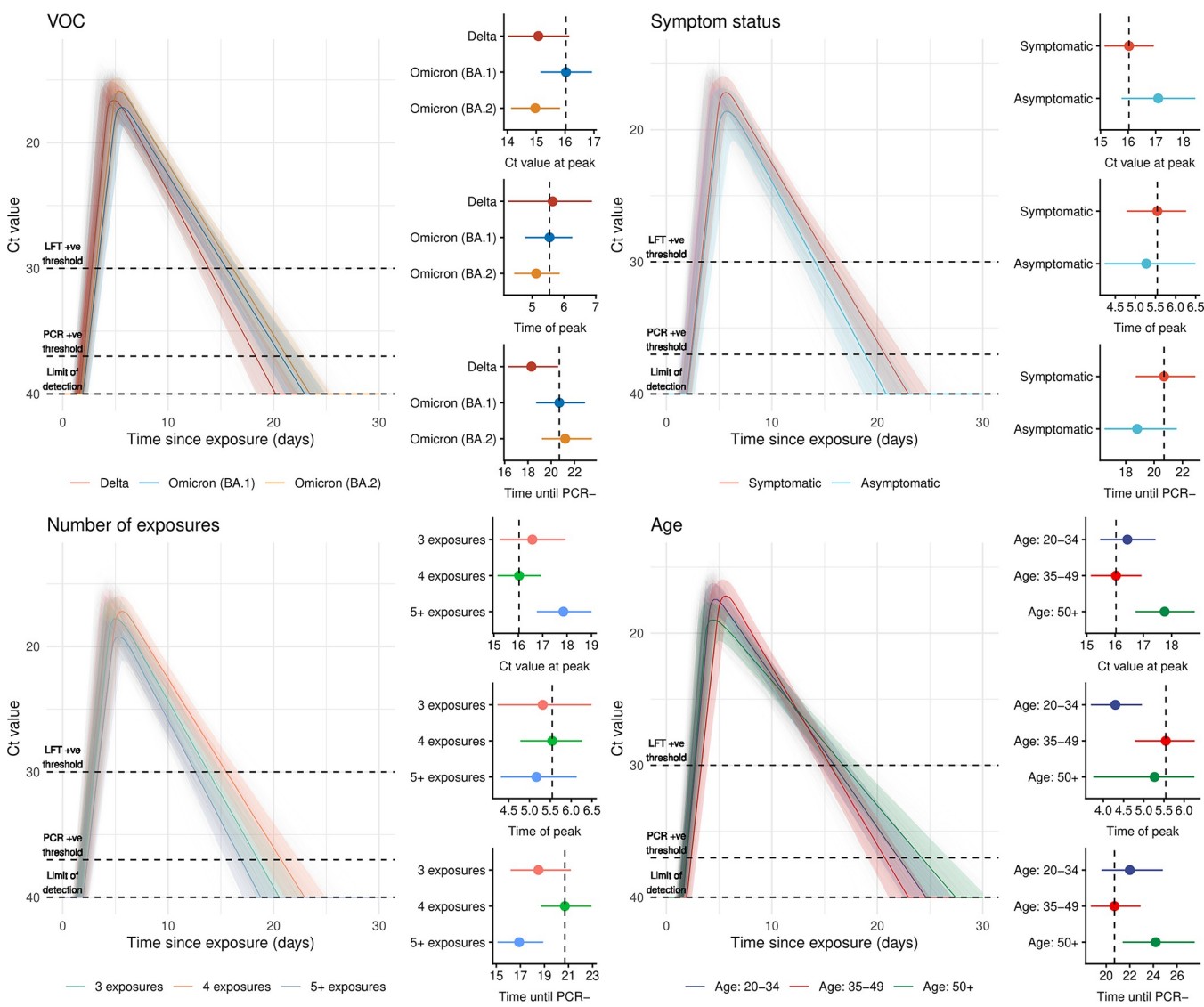

**Fig 3. Fitted covariate-level Ct trajectories and the differences in the parameter values stratified for each covariate. All left panels**: Model fits of the viral kinetics over the course of whole infections, beginning from the inferred exposure time. After fitting the model, we simulated 10,000 trajectories, using parameters sampled from the inferred posterior distributions for each covariate considered. We then calculate the median and 95% CrI for each covariate and plot the resulting trajectories as coloured regions, with a sample of 100 trajectories plotted transparently beneath. **All right panels**: The median and 95% estimates of the inferred parameter values for the peak Ct value, the timing of the peak, and the time at which the trajectory hits the LOD. Dashed line shows the value of the baseline participants inferred parameter values. **(A)** Trajectories and parameter values for the VOCs considered. **(B)** Trajectories and parameter values for the symptom statuses considered. **(C)** Trajectories and parameter values for the numbers of exposures considered. **(D)** Trajectories and parameter values for the age groups considered. Data underlying the graphs in this figure can be found in https://zenodo.org/doi/10.5281/zenodo.10223385. CrI, credible interval; Ct, cycle threshold; LOD, limit of detection; VOC, variant of concern.

estimates of 17.7 (95% CrI: 16.4 to 18.9), 1.8 (95% CrI: 0.5 to 3.2) Ct values higher than baseline individuals (**Fig 3D** and **S6 Table**).

## Differences in timing by variant and host characteristics

We estimated that the Ct trajectories in the baseline group (BA.1 infections in 35 to 49 years old after 3 vaccinations) peaked at 5.9 days after exposure (95% CrI: 5.2 to 6.7) (**Fig 3A** and **S2 Table**). Delta Ct trajectories peaked at 6.6 days after exposure (95% CrI: 5.1 to 8.3), and BA.2

trajectories peaked at 5.2 days after exposure (95% CrI: 4.3 to 6.1), 0.53 (95% CrI: −0.88 to 2.3) days later, and 0.74 (95% CrI: −1.7 to 0.25) days sooner than the baseline participants, respectively (**Fig 3A** and **S3 Table**). Trajectories of asymptomatic participants peaked 5.5 (95% CrI: 4.2 to 6.9) days after exposure, 0.39 (95% CrI: −1.8 to 1.1) days sooner than participants in the baseline group (**Fig 3B** and **S4 Table**). We did not find a substantial difference in the timing of the peak for differences in the total numbers of exposures (**Fig 3C** and **S5 Table**) or for 50 + year-old individuals (**Fig 3D** and **S6 Table**). Lastly, we found that Ct trajectories for 20- to 34-year-old individuals peak sooner than the baseline group, at 4.5 (95% CrI: 3.7 to 5.4), 1.4 (95% CrI: 0.42 to 2.2) days sooner than baseline individuals (**Fig 3D** and **S6 Table**).

We then estimated the times at which Ct trajectories reached the thresholds of Ct = 37 or Ct = 30, approximating the LODs by PCR or a rapid test that can only detect high Ct values. For PCR positivity (PCR+), baseline trajectories crossed the PCR+ threshold at 2.6 (95% CrI: 1.7 to 3.3) days after exposure and would have remained PCR+ until 23.8 (95% CrI: 20.7 to 26.2) days after exposure. We estimated that Delta trajectories would turn PCR+ at an almost identical time to the baseline trajectories but would only remain PCR+ until 21.5 (95% CrI: 18.1 to 24.2) days after exposure. We estimated that BA.2 infections would turn PCR+ 2.2 (95% CrI: 1.4 to 2.9) days after exposure and would remain PCR+ until 25.0 (95% CrI: 21.3 to 27.8) days after exposure. Trajectories for all other covariates were estimated to turn PCR+ at very similar times to the baseline trajectories. However, we estimated substantial variation in the duration of PCR positivity for the other covariate categories. Trajectories would become PCR− for participants with 3 or 5+ total antigenic exposures sooner than baseline individuals; specifically at 21.4 (95% CrI: 17.4 to 25.1) and 19.1 (95% CrI: 16.1 to 21.4) days after exposure respectively. Lastly, we found that trajectories for participants in age groups 20 to 34 and 50+ would become PCR − later than baseline individuals, at 24.6 (95% CrI: 21.0 to 27.7) and 26.9 (95% CrI: 22.6 to 30.0) days after exposure, respectively. Estimated rapid test positivity timings are reported in **S1 Text**.

### Summary of estimated variations in Ct dynamics by variant and host characteristics

Differences within and between all 4 covariate categories were difficult to characterise precisely, given the complex nature of the exposure history of this dataset. However, some patterns emerged. To summarise, compared to reference individuals, we found lower peak Ct values for Delta- and BA.2-infected individuals; a later peak timing for Delta-infected individuals and an earlier peak timing for BA.2-infected individuals; Delta-infected individuals reaching the LOD sooner than BA.1- or BA.2-infected individuals; a substantially higher peak Ct value for individuals with 5+ exposures—compared to reference individuals and individuals with 3 exposures—consistent with lower viral loads and higher levels of immunity for those individuals [20]; higher peak Ct values for asymptomatic individuals, consistent with previous estimates and consistent with lower viral loads in asymptomatic individuals throughout their infections [7,25]; and, lastly, the timing of the peak Ct value occurred sooner for asymptomatic individuals, consistent with previous estimates [7]. However, we estimated that peak Ct values, timing of peak, and LOD did not follow a consistent pattern in our dataset across the different age groups. This could be due in part to low statistical power for the highest age group and/or the lack of explicit inclusion of complex underlying antibody dynamics in our model.

### Estimating the relationship between timing of peak shedding and symptom onset

Combining the fitted Ct trajectories with incubation period estimates, we estimated the temporal dynamics of viral shedding and their relation to the timing of symptom onset for each

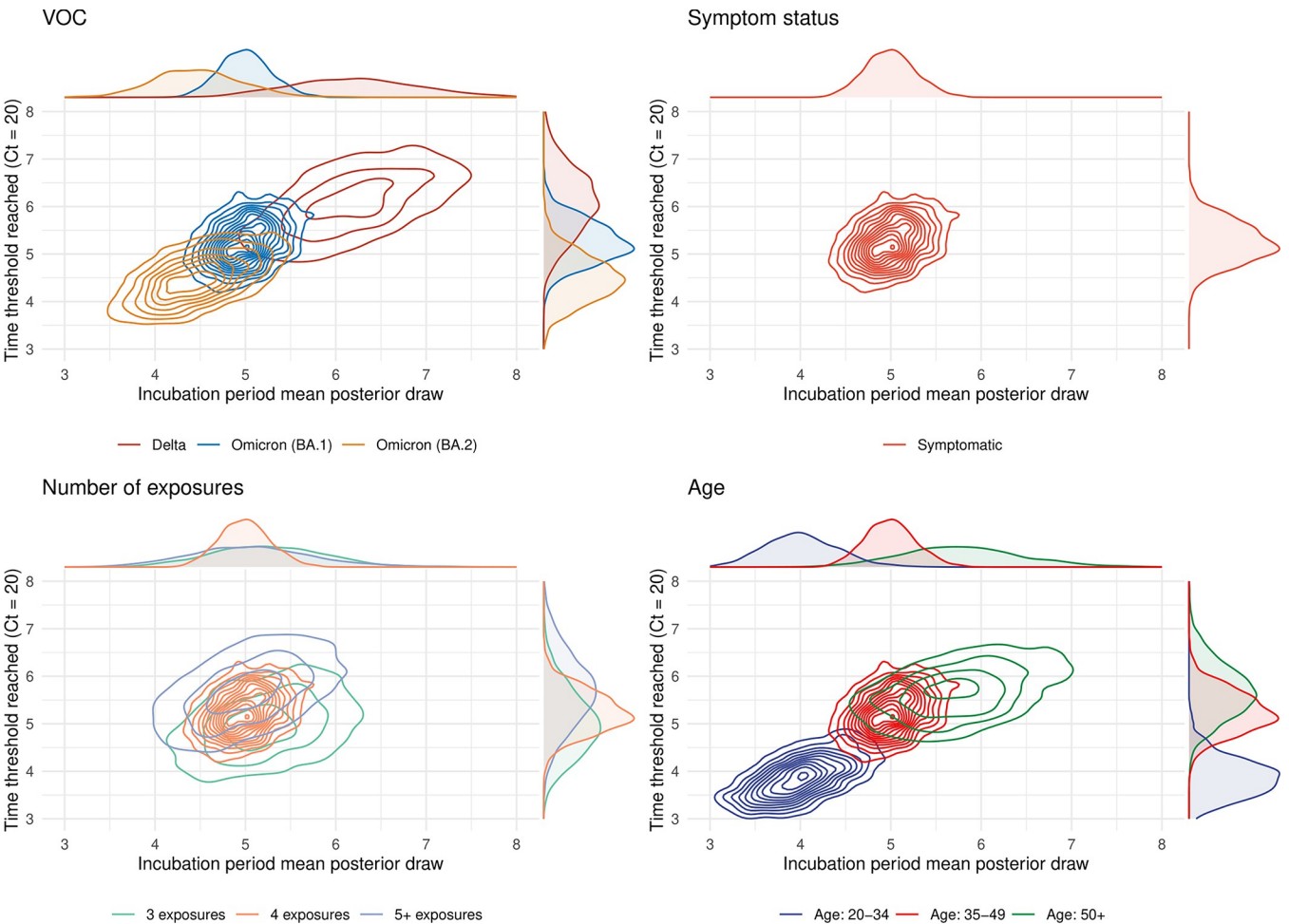

**Fig 4. Bivariate density plots of median incubation periods against the time at which trajectories surpassed an assumed Ct value corresponding to high viral loads, for each covariate. All panels**: We plot the posterior distributions for the estimated incubation periods, stratified by all covariates considered, against the time at which the simulated Ct trajectories, as shown in Fig 3, cross an assumed Ct value threshold for the first time (Ct value = 20). This Ct value represents a time point in the covariate-level trajectories of high viral load. **(A)** Parameter values for the 3 VOCs considered. **(B)** Parameter values corresponding to symptomatic infections. **(C)** Parameter values corresponding to the numbers of exposures considered. **(D)** Parameter values for the age groups considered. An alternative Ct value threshold (Ct = 25) was considered in an analogous analysis in the Supporting information (**Fig N in S1 Text**). Data underlying the graphs in this figure can be found in https://zenodo.org/doi/10.5281/zenodo.10223385.

covariate. To compare dynamics, we selected a Ct threshold relating to a time point corresponding to comparatively high viral load (Ct = 20). We found that individuals infected with BA.2 reach the assumed Ct threshold on their upwards trajectory earlier than those infected with BA.1 and that BA.1 individuals reached the Ct threshold earlier than Delta-infected individuals. Similarly, using our incubation period estimates, we estimated that symptom onset was earlier for BA.2-infected individuals than for BA.1-infected individuals, and earlier for BA.1-infected individuals than for Delta-infected individuals (**Fig 4**).

Estimates stratified by VOC revealed that these 2 timescales have become shorter as variants have emerged (**Fig 4A**). For example, the marginal estimates for the time at which the Ct trajectories reached the assumed threshold move from 5.6 (95%: 3.8 to 7.3) days to 5.2 (95%: 4.1 to 6.2) days to 4.5 (95%: 3.3 to 5.5) for the VOCs in the order in which they evolved: Delta ➔ BA.1 ➔ BA.2. Similarly, the incubation periods are 6.0 (95%: 4.2 to 7.5) days to 5.0 (95%: 4.3 to

5.6) days to 4.5 (95%: 3.3 to 5.5) days for the same ordering (**Fig 4A**). As asymptomatic individuals do not have an incubation period, such estimates by symptom status were only possible for symptomatic individuals, which were used as the baseline set of individuals and therefore match those of the BA.1 estimates (**Fig 4B**). We did not find substantial deviations from the reference group when investigating this relationship by total numbers of antigenic exposures (**Fig 4C**). Lastly, when stratified by age group, we estimated the time trajectories pass the assumed threshold moves from 5.7 (95%: 4.3 to 7.0) days to 5.2 (95%: 4.1 to 6.2) days to 4.0 (95%: 2.9 to 5.0) days, and the onset of symptoms occurs at 5.8 (95%: 4.0 to 7.2) days to 5.0 (95%: 4.3 to 5.6) to 4.1 (95%: 3.0 to 5.0) days when age groups are reported by decreasing order: 50+ ➔ 35 to 49 ➔ 20 to 34 (**Fig 4D**). Our estimated incubation periods, sampled using both the inferred mean and standard deviation parameters, are included in the Supporting information (**Fig O in S1 Text**).

## Sensitivity analyses and robustness checks

To test the sensitivity of our model to the choice of semi-informative priors, the sample sizes, and our choice of covariates, we performed a number of additional robustness checks, sensitivity analyses, and a model selection procedure.

First, to investigate sensitivity to the semi-informative prior choices, we widened the covariate-level priors by increasing the variance parameter considerably for the 3 main process parameters: Ct value at peak, timing of the peak, and timing the LOD is reached. We still used normal distributions for all covariate-level priors, as they are highly recommended for high-dimensional hierarchical models such as this one, to aid with sampling from complex geometries. However, where the variance parameter was between 0.5 and 1 for all semi-informative priors—with mean values chosen from literature—we multiplied all variance parameters by 5, widening the priors considerably. The parameters are on logit (Ct value parameters) or log scales (timing parameters), meaning that this increase in the variance, when measured in natural units (Ct values or time (days)), should also be interpreted on these same scales. The results using less informative priors are in line with the semi-informative priors (**Fig T in S1 Text**). The direction of all effect sizes is the same between the two, and all magnitudes are within each other's credible intervals. The overall uncertainty for the less informative prior case is slightly wider but at a level hard to distinguish visually. We conclude from this that our results from our main analysis using semi-informative priors are not overly sensitive to the choices made and that our resulting estimates are mostly informed by our modelling framework conditioned on our dataset.

Second, to test the ability of our model to recover known parameter values, we performed a simulation study (described in detail in **S1 Text**). **Figs Y-Z in S1 Text** show results after fitting our model to simulated data whereby we demonstrate the ability of our modelling framework to recover known effect sizes varying 3 key process parameters between 2 synthetic groups smaller than or similar in size to the smallest group of individuals in our analysis.

Third, we performed a number of model fits with different sets of covariates and prior choices, all of which are reported in the Supporting information (**Figs P-U in S1 Text**). Specific results and their interpretation are described in the Supporting information. However, to summarise, we found the results from the various additional checks were in line with the main results (**Figs 3 and 4**).

Lastly, **S7 Table** shows the results from an approximate leave-one-out cross-validation performed for 7 candidate models, finding similar predictive power between models with different design matrices, justifying the use of the specific set of covariates used in the main analysis.

## Discussion

We estimated differences in viral shedding dynamics over the course of infections with different SARS-CoV-2 VOCs by combining data from a large cohort of healthy adults with a statistical model that accounted for both individual- and group-level variation in Ct value. Adjusting for a changing background of host characteristics over time, we identified differences in Ct kinetics that were associated specifically with VOC status and found that Delta and BA.2 infections tended to peak at higher viral loads compared to BA.1 infections. We also found that age and number of prior exposures had a strong influence on Ct peak, with lower shedding among older individuals and those who had at least 5 exposures to vaccination and/or infection. Moreover, we found evidence of a correlation between the speed of early shedding and duration of incubation period when comparing different VOCs and age groups.

The unique dataset and model developed here allow us to expand upon previous findings from recent studies that investigated specific aspects of viral kinetics of SARS-CoV-2 infections, both in a human challenge model [4] and in the community [5,7,11–14,19,25]. First, we collected serial swabs from participants in a large, well-defined cohort, which is broadly representative of the working-age UK population and who were undergoing regular surveillance RT-PCR testing. Second, we captured both asymptomatic and symptomatic infections with high confidence, linked to the last negative test, in comparison with other studies [25] that relied upon self-initiated symptom-based testing. This knowledge of prior negative test timings allowed us to estimate the duration of presymptomatic infection with greater confidence and directly compare the peak and trajectory of symptomatic and asymptomatic infections. One of the major limitations of most studies of SARS-CoV-2 infection has been the unknown time between infection and symptom onset. While the human challenge model approach has provided detailed information about viral kinetics following inoculation, participants were unvaccinated, healthy younger adults infected with a small dose of ancestral "pre-VOC" virus [4], limiting inferences about the kinetics of emergent VOCs within the majority-vaccinated general public. Using the last known negative test in our participants, through to their first positive test, and then through serial tests up to 30 days afterwards, we were able to fully characterise viral kinetics over the course of infection rather than focusing exclusively on the initial post-onset period [11,12]. Furthermore, by linking to dates of symptom onset and conditioning on each participant's estimated exposure time, we jointly—alongside each individual's viral kinetics—estimated an incubation period for each covariate. Finally, we included participants with Delta, Omicron BA.1, and Omicron BA.2 infections, enabling direct comparison between VOCs that have circulated since May 2021 in the United Kingdom [15].

Our analysis made it possible to evaluate several putative relationships between host characteristics, disease profile, infecting variant, and viral kinetics. First, it has previously been suggested that both vaccination and milder or asymptomatic infection may be associated with lower viral loads and faster viral clearance [12,26]. Our finding of higher Ct values after 5 + viral exposures (whether vaccine or infection) is consistent with enhanced host immune responses due to previous exposure to spike protein. Among asymptomatic infections, there was some—albeit weak—evidence pointing to lower and earlier peak viral load. We did not find evidence that people with asymptomatic infections cleared the virus more quickly, with predicted Ct remaining below 30 for 14 days after exposure. These findings are consistent with those in the human challenge model, where viral loads remained high regardless of symptom severity [4]. Within our cohort, effects of vaccine doses or symptom status on the value or timing of the peak Ct value were small, suggesting that the impact of isolation periods or likelihood of transmission would not differ based on presence of symptoms or number of previous

exposures. However, we did not have data from participants with no prior exposures, limiting our ability to fully explore the role of heterogeneity in immunity.

Second, while a report on a large community-based study reported that Omicron BA.1 caused shorter symptomatic illness compared to the Delta variant [27], our estimated viral load kinetics did not indicate faster clearing of Omicron BA.1. This is consistent with our previously reported data on symptom severity and duration, which found these to be the same across all VOC infections considered here [22]. It has been suggested that different VOCs have altered tropism for higher or lower portions of the respiratory tract [4,28], with BA.1 preferring upper portions and Delta those deeper. These differences in anatomical preference may alter Ct dynamics in the nasopharynx, and, hence, the symptom constellation may also be different with coryzal symptoms favoured in BA.1 and BA.2 infections compared to Delta [22,29]. We found a clear difference in time to LOD for Delta infections compared to Omicron BA.1 infections (in 35 to 49 yeas old with 4 previous exposures), but the majority of people had recently received their third vaccine dose at the time of infection and were more likely to be asymptomatic, suggesting a role of faster viral clearance for the nasopharynx in the immediate post-vaccine period [4,30]. We also found evidence of an earlier peak Ct and higher peak viral load in BA.2 infections, consistent with a recent study of serial viral loads by [27,31] and the observed rapid spread of BA.2 across the globe during the latter stages of the BA.1 epidemic [32].

Third, the arrival of each new VOC in the UK typically resulted in a wave of infections concentrated in specific age groups, while at the same time vaccine doses have been prioritised towards older age groups [15,33]. We found the peak Ct was lower in younger adults across all VOC, with older adults peaking later, and having a shorter trajectory to the LOD, despite similar vaccination doses at the time of infection. Indeed, the older adults, who received vaccine doses significantly earlier than the younger cohort, had less viral shedding. While not formally included in the model, older adults typically have an attenuated neutralising antibody response after COVID-19 vaccination [20,21,34]. One might expect in this scenario that viral replication would therefore be more active and prolonged. Further study to understand why we observed the opposite is needed.

There are some additional limitations in our analysis. Error in Ct values is known to decrease at higher viral loads and increase at lower viral loads [11] (**Fig B in S1 Text**), but to ensure model identifiability, we assumed a constant measurement error across the range of reported Ct values. Such an approximation would be particularly important to revisit if we were analysing timescales beyond the relatively short postinfection period considered here and, hence, a larger range of Ct values. We also used prior exposures as a proxy to capture the potential effects of existing immunity. With more detailed linked serological and PCR data, it would be possible to examine this relationship in more detail, although this would require a substantially more complex model structure. Our finding of higher Ct values after 5+ viral exposures (whether vaccine or infection) is consistent with enhanced host immune responses due to previous exposure to spike protein. However, the nonincreasing relationship in Ct estimated between 3 and 4 exposures indicates that immunological dynamics, particularly following initial vaccinations and infections at different points in time, may be more complex than can be fully captured in a single exposure total.

To test the relative predictive performance of several model candidates, we performed a leave-one-out (LOO) model selection procedure for 8 candidate models (see **S7 Table** for the candidate models considered). The candidate models have either different design matrices, specifying which set of covariates are included in each model, or different priors to the model used in the main analysis. The predictive power of all models included in the LOO analysis with different design matrices are within 4 standard errors of one another, which is broadly

the rule of thumb for a model with significantly lower predictive power than another. Therefore, we can conclude that including covariates known to influence viral kinetics is justified, given that the models have similar predictive power with or without their inclusion. The model with noninformative priors, which has the lowest predictive power compared to all other candidates—relative to the model with semi-informative priors used in the main analysis (**S7 Table**)—is still within the range where it is difficult to conclude an overall difference in predictive power. As such, we also performed a complementary sensitivity analysis in which we plot the effect sizes of the model with noninformative priors (**Fig T in S1 Text**). The central effect size estimates are broadly similar in both the semi- and noninformative models (**Fig 3 and Fig T in S1 Text**). However, the resulting uncertainty is higher in the noninformative prior case. Combining the interpretation from the LOO and sensitivity analysis around the influence of semi-informative priors, we conclude that their inclusion in the results reported in the main analysis is justified, given that the predictive power is higher with them, the central estimates are broadly similar, and prior information is only used where reliably reported elsewhere in published literature.

Previous studies inferring viral kinetics have concentrated mostly on differences between VOCs circulating at the time in question. As such, they did not consider the host of other possible confounding variables, making their results difficult to attribute solely to differences in infecting VOC [5,11,30,35]. In interpreting effect sizes between viral kinetic estimates, we controlled for the complex emerging picture around individual life-course exposures to SARS-CoV-2, both as infections and vaccines. Differences in numbers of infections, the VOC causing the infection, the numbers of vaccinations, and many other individual-level variables introduce many sources of possible confounding. In particular, the timing of the sampling within our study differs compared to other studies, and, hence, Ct dynamics by variant could vary substantially across populations if these factors are not adjusted for.

Moreover, we find less difference in viral kinetics by age in comparison to other studies [5,7]. Again, this is likely to be influenced by a number of factors. Given the complex relationship between age, symptom severity, immunity, and exposure risk, studies that do not account for these characteristics could attribute differences in shedding to age, rather than another factor that correlates with age. However, combining the underlying viral kinetics model with a Bayesian hierarchical framework allowed such information to be jointly incorporated into our analysis. Additional factors that could influence observed differences in effect size estimates include calibration differences between PCR pipelines resulting in differences in the absolute Ct values; differences in sampling procedure with previous studies primarily relying on symptom-triggered sampling; complex exposure histories following both the vaccine rollout and the emergence of new VOCs; and the ability of the underlying model structure to account for the numerous covariates suspected to affect viral kinetics or the inherent individual-level heterogeneity in response to exposure to SARS-CoV-2. We combine data from a study designed to capture viral load across entire infections, including sampling before and after exposure, with a model structure complex enough to adjust for all necessary covariates and individual-level variation in viral kinetics and able to inform model parameters by jointly fitting to data where possible. Therefore, the differences between our effect sizes and previously published studies may be explained by the combined adjustment for all of these complexities.

The flexibility provided by our framework provides a methodological foundation for future work using similar datasets. Given the ongoing antigenic turnover of SARS-CoV-2 variants of concern since late 2020, our results show that it will be increasingly important to adjust for a range of changing host factors when quantifying viral kinetics and considering implications for potential infectiousness in future.

## Supporting information

**S1 Table. Table of summary statistics for all infection episodes for infection-naïve individuals.** We present a table of summary statistics, including the number of points available per infection episode, the delay between first viral load data and symptom onset, the observed peak viral load, and the time until clearance (approximated by the final time point per infection episode) for all individuals that had not had an infection before joining the study ($N = 75$).
(DOCX)

**S2 Table. Table of descriptions of each model parameter and priors.** We list each parameter within our modelling framework, its biological interpretation, the prior distribution we chose for it, and any references justifying the choice of prior.
(DOCX)

**S3 Table. Table of the median and 95% credible intervals of the population-level Ct model parameters by VOC.**
(DOCX)

**S4 Table. Table of the median and 95% credible intervals of the population-level Ct model parameters by symptom status.**
(DOCX)

**S5 Table. Table of the median and 95% credible intervals of the population-level Ct model parameters by total number of antigenic exposures.**
(DOCX)

**S6 Table. Table of the median and 95% credible intervals of the population-level Ct model parameters by age.**
(DOCX)

**S7 Table. Full table of ELDP, ELDP differences, standard errors, and SE differences, as computed using the loo function incorporated in the cmdstanr R package.** The models are reported in no particular order.
(DOCX)

**S1 Text. Supporting material file.** Full supporting information, including all supporting figures and detailed methods. Supporting tables are included as separate files.
(PDF)

## Acknowledgments

The authors would like to thank all the study participants, the staff of the NIHR Clinical Research Facility at UCLH including Miguel Alvarez and Marivic Ricamara. We would like to thank Bobbi Clayton, Gita Mistry, Simon Caiden, and the staff of the Scientific Technology Platforms (STPs) and COVID-19 testing pipeline at the Francis Crick Institute. We thank Prof. Wendy Barclay of Imperial College and the wider Genotype to Phenotype consortium for the Alpha and Delta strains used in this study, and Max Whiteley and Thushan I de Silva at The University of Sheffield and Sheffield Teaching Hospitals NHS Foundation Trust for providing source material. We thank Prof. Gavin Screaton of the University of Oxford for the Omicron BA.1 strain used in this study.

## Author Contributions

**Conceptualization:** Timothy W. Russell, Sam Abbott, Edward J. Carr, Lloyd A. C. Chapman, Rachael Pung, Billy J. Quilty, David Hodgson, Rupert Beale, David L. V. Bauer, Emma C. Wall, Adam J. Kucharski.

**Data curation:** Hermaleigh Townsley, Sam Abbott, Joel Hellewell, Edward J. Carr, Ashley S. Fowler, Lorin Adams, Chris Bailey, Harriet V. Mears, Ruth Harvey, Bobbi Clayton, Nicola O'Reilly, Yenting Ngai, Jerome Nicod, Steve Gamblin, Bryan Williams, Sonia Gandhi, Charles Swanton, David L. V. Bauer, Emma C. Wall.

**Formal analysis:** Timothy W. Russell, Sam Abbott, Joel Hellewell, Lloyd A. C. Chapman, Adam J. Kucharski.

**Funding acquisition:** Charles Swanton, Emma C. Wall.

**Investigation:** Charles Swanton, Rupert Beale, David L. V. Bauer, Emma C. Wall, Adam J. Kucharski.

**Methodology:** Timothy W. Russell, Hermaleigh Townsley, Sam Abbott, Joel Hellewell, Edward J. Carr, Lloyd A. C. Chapman, Rupert Beale, David L. V. Bauer, Emma C. Wall, Adam J. Kucharski.

**Project administration:** Rupert Beale, David L. V. Bauer, Emma C. Wall, Adam J. Kucharski.

**Software:** Sam Abbott.

**Supervision:** David L. V. Bauer, Emma C. Wall, Adam J. Kucharski.

**Validation:** Timothy W. Russell, Hermaleigh Townsley, Sam Abbott.

**Visualization:** Timothy W. Russell, Sam Abbott, Joel Hellewell, Edward J. Carr, Rachael Pung, David L. V. Bauer, Emma C. Wall, Adam J. Kucharski.

**Writing – original draft:** Timothy W. Russell, Hermaleigh Townsley, Sam Abbott, Joel Hellewell, Edward J. Carr, Rupert Beale, David L. V. Bauer, Emma C. Wall, Adam J. Kucharski.

**Writing – review & editing:** Timothy W. Russell, Hermaleigh Townsley, Sam Abbott, Joel Hellewell, Edward J. Carr, Lloyd A. C. Chapman, Rachael Pung, Billy J. Quilty, David Hodgson, Ashley S. Fowler, Lorin Adams, Chris Bailey, Harriet V. Mears, Ruth Harvey, Bobbi Clayton, Nicola O'Reilly, Yenting Ngai, Jerome Nicod, Steve Gamblin, Bryan Williams, Sonia Gandhi, Charles Swanton, Rupert Beale, David L. V. Bauer, Emma C. Wall, Adam J. Kucharski.

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
