## [Editor Report · Decision Letter 0]

6 Apr 2023

Dear Dr. Russell, 

Thank you for submitting your manuscript entitled "Within-host SARS-CoV-2 viral kinetics informed by complex life course exposures reveals different intrinsic properties of Omicron and Delta variants" for consideration as a Research Article by PLOS Biology.

Your manuscript has now been evaluated by the PLOS Biology editorial staff and I am writing to let you know that we would like to send your submission out for external peer review. Please note, however, that the outcome of our discussion of your manuscript is that we have some reservations as to level of conceptual advance with respect to previous studies. We would need to be persuaded by the reviewers that the paper has the potential after revision to offer the significant strength of advance that we require for publication in order to pursue it further for PLOS Biology.

Before we can send your manuscript to reviewers, we need you to complete your submission by providing the metadata that is required for full assessment. To this end, please login to Editorial Manager where you will find the paper in the 'Submissions Needing Revisions' folder on your homepage. Please click 'Revise Submission' from the Action Links and complete all additional questions in the submission questionnaire.

Once your full submission is complete, your paper will undergo a series of checks in preparation for peer review. After your manuscript has passed the checks it will be sent out for review. To provide the metadata for your submission, please Login to Editorial Manager (https://www.editorialmanager.com/pbiology) within two working days, i.e. by Apr 08 2023 11:59PM.

Kind regards,

Paula

---

Senior Editor

PLOS Biology

---

## [Decision Letter · Decision Letter 1]

22 Jun 2023

Dear Dr. Russell,

Thank you for your patience while your manuscript "Within-host SARS-CoV-2 viral kinetics informed by complex life course exposures reveals different intrinsic properties of Omicron and Delta variants" was peer-reviewed at PLOS Biology. It has now been evaluated by the PLOS Biology editors, an Academic Editor with relevant expertise, and by several independent reviewers. 

In light of the reviews, which you will find at the end of this email, we would like to invite you to revise the work to thoroughly address the reviewers' reports.

As you will see below, the reviewers think that the manuscript is interesting but they find some issues related with your data analyses and interpretation of the results that should be solved before publication. We think that a clearer exposition of the statistical analysis would benefit the paper. In particular, we agree with the reviewers that you should double check for the significance of the variant effects by testing your models against models without variants as a cofactor, and also to discuss the estimated effects sizes in comparison with previous estimates. This is particularly relevant as the difference between Delta and Omicron variants are highlighted in the title, while the abstract highlights mostly effects of age and prior exposure.

Given the extent of revision needed, we cannot make a decision about publication until we have seen the revised manuscript and your response to the reviewers' comments. Your revised manuscript is likely to be sent for further evaluation by all or a subset of the reviewers.

**IMPORTANT - SUBMITTING YOUR REVISION**

*Re-submission Checklist*

*Published Peer Review*

*PLOS Data Policy*

*Blot and Gel Data Policy*

Sincerely,

Paula

---

Senior Editor

PLOS Biology

REVIEWS:

Reviewer #1: Mathematical modelling of viral dynamics.

Reviewer #2: Infectious diseases dynamics and transmission.

Reviewer #1: The study by Russel et al. proposes an analysis of viral kinetics in individuals infected by Omicron or Delta variants. Despite the millions of people that have been infected by these viruses, the specific patterns of viral kinetics associated with these infections remain obscure. This is due to the fact that analyzing retrospective cohorts of infected individuals, typically sampled after symptom onset and when viral load is already in its clearance phase, is prone to many biases and confounding factors: immunity levels, vaccination rate, incubation period, testing policy…

Therefore, the only study design to properly address this question is to constitute a cohort of uninfected individuals followed longitudinally, frequently tested to detect early infection, and in whom positive PCR triggers and even more frequent sampling of viral load. This is exactly what the authors have designed, at least in a subset of their population, and this is what makes this study novel and interesting. Such design is however expensive and difficult to implement, relying on a less captive population of outpatients, and the data presented, albeit being very original, remain limited to address the complex question of viral kinetics. Overall the effect size that are found seem very limited, suggesting no major differences across variants. As such some of the results raise concerns about the possibility to really extrapolate from this population, and I am not totally convinced that the studied population is sufficiently large to really disentangle the different confounding factors. 

I have the following comments: 

* It is nice to have conceptual figures but figure 2 is probably too complex. The originality of the study lies in the population of individuals that joined before infection episode (N=75) and I would suggest to add simple information focusing on this population as follows: 

o Summary statistics: number of points available per patient, delay between first viral load data and symptom onset, observed peak viral load, time to clearance and basic statistic comparisons between the 3 viral strains. Even if there are a number of confounding (that will be addressed by modeling) please show viral load data available at different time points for each variant and basic summary statsitics to compare variants. If data are sparse, you can for instance bin timepoints (every two of three days). The time reference could be time since first positive PCR. This would help appreciate the quantity of info available (and probably would be more informative than the color code used in figure 2).

* I am fine with Bayesian analyses but again very difficult to appreciate to what extent results are sensitive to prior distributions. Please provide the prior distributions that were used, and in case they are "semi-informative" I would recommend to conduct a sensitivity analysis to evaluate to which extent results were dependent on the prior, and what would be the results in case of uninformative priors. This would help modelers appreciate what can be reliably inferred from these data, and what requires to borrow from literature. 

* Some of the results seem in contradiction with the literature which raise concern about the extrapolation that can be made from the data

o There is large body of data suggesting that age is associated with extended time to viral clearance. I find it very surprising that age groups 20-34 and 50+ seem to have similar time to viral clearance. It is possible that this is due to the fact that the model accounts for repeated exposure, which is not done usually but still seems very counterintuitive. See for instance model based approaches in Hay et al, Elife (2022); Neant et al., PNAS (2021) 

o Same thing for the peak viral load. It was my understanding that delta was associated with higher peak viral load than BA.1. Perhaps not in the order published initially (*1000) but not as small as suggested here (1 Ct, eg, 2 fold). For instance in Hay et al, the difference seems to be ~1 log (Table S5). In the macaque model (https://www.biorxiv.org/content/10.1101/2022.11.09.515748v1), the difference was also larger (close to 2 logs, Table S1). 

I understand the difficulty to arrive to a consensus value given all the confounders, but I think that the authors should acknowledge that some of these results are not necessarily consensual, and should discuss what could explain the discrepancy between their results and those of the literature. Going back to my first point, perhaps it would also be nice to discus the differences between the model-based approach and the crude summary statsitics of the data. It is possible that the authors show that it is precisely the model based approach that allows to adjust some metrics and could explain some of the discrepancy with more descriptive approaches. 

Reviewer #2: In this manuscript, the authors have fitted a mathematical model of viral kinetics for SARS-CoV-2 kinetics, which they have fitted to data from an observational cohort study of health adult volunteers. This is a very interesting piece of work: the dataset is detailed and complex, and the modelling has been developed and carried out in a way that reflects this. In general, the methods and results are well presented. I have a few concerns and questions, especially around interpretations of the results, which should be addressed. Nevertheless, I commend the authors on an interesting study.

* Personally, I wouldn't describe the model as mechanistic. The viral load model is a piece-wise regression model: no mechanism is proposed for why the exponential growth of the virus (linear decrease of the Ct values) stops (resource depletion, immune response, etc.), and viral load then declines. However, these terms are used differently in different fields: I'll leave the final choice of this to the authors.

* I'm a little unclear on the definition of the incubation period: in Figure 1B, it appears to be the time between exposure and the peak viral load. However, lines 182-186 in the manuscript suggest it is the time different between exposure and symptom onset. Can this be clarified?

* I'm a bit confused about the various censoring used in the model likelihood. There is a truncation below Ct=0: presumably a PCR machine is incapable of recording a negative number of cycle thresholds? I would say that the likelihood should instead be truncated at zero, but I'm not sure this is an important distinction. Based upon figures like Fig S6, it looks like very few (if any) datapoints show Ct<10, so I don't think this is influencing the model results. The censoring at Ct>40 is a more important, however, and has confused me. It is unclear to me if this represents the limit of detection (LOD) of the PCR machine(s) used in this study, or whether it is simply the smallest Ct value recorded in the dataset? The latter might explain why the authors are also trying to infer an additional LOD parameter. Additional clarification is needed here.

* I found the description of the hierarchical model hard to follow. Line 127 of the S.I. presents the Ct kinetics model, with 6 parameters- possibly these are population-level averages of these parameters, which can vary between individuals? Lines 187-192 give priors for 'covariate-level parameters', although it is not stated what these covariates could be. From the presentation of the results in the figures, I guess these could be things like: age group, number of prior exposures, VOC, etc. Nevertheless, this presentation is quite shoddy. Then lines 227-234 give the individual-level parameters. Would it be possible to give an example of the full model equation for a particular individual, 'i', in a particular age group, with a given number of exposures, infected with a particular variant, with the individual-level variation included? Or are the covariates fitted one at a time in the models? 

* Regarding the results presented for things like 'differences in peak Ct value by variant', the authors say very little about the significance (I use the word broadly, rather than in a statistical sense) of their results. Differences (with 95% credible intervals) are presented for the peak Ct value in the (e.g.) Delta group compared to the Omicron BA. 1. I suspect the authors consider that if the 95% CrIs don't cross 0 then the difference should be considered 'meaningful'? Is there any reason why the authors didn't do a more formal analysis e.g. with WAIC or LOO-CV? In other words, would a simpler model, without this covariate for variant in the model, explain the data more poorly? The authors may have good reason for not proceeding in this manner, but I'd be interested to hear their justification

* Related to the above point, on line 279 the authors say that individuals with 3 or 4 exposures have 'substantially' higher viral load, compared to individuals with 5 exposures. On what basis is the word 'substantially' used? The 95% CrI don't seem to support this, if it means that the difference is significant/meaningful. Furthermore, Figure 3C suggests a non-monotonic relationship between the peak Ct value and the number of prior exposures, so this situation seems muddier to me than the authors indicate.

* Line 408: I'm not sure I would describe the cohort as 'broadly representative of the UK population', as it seems to solely consist of working age people. It might be reasonable to say 'broadly representative of the working-age UK population'.

* Supporting Information, line 139: this equation is missing a closing bracket

---

## [Decision Letter · Decision Letter 2]

6 Nov 2023

Dear Dr. Russell,

Thank you for your patience while we considered your revised manuscript "Within-host SARS-CoV-2 viral kinetics informed by complex life course exposures reveals different intrinsic properties of Omicron and Delta variants" for publication as a Research Article at PLOS Biology. This revised version of your manuscript has been evaluated by the PLOS Biology editors, the Academic Editor and the original reviewers.

Based on the reviews, we are likely to accept this manuscript for publication, provided you satisfactorily address the remaining points raised by the reviewers. Please also make sure to address the following data and other policy-related requests.

1. ETHICS STATEMENT:

Please include information about the form of consent (written/oral) given for research involving human participants. All research involving human participants must have been approved by the authors' Institutional Review Board (IRB) or an equivalent committee, and all clinical investigation must have been conducted according to the principles expressed in the Declaration of Helsinki.

2. DATA POLICY:

A) Supplementary files (e.g., excel). Please ensure that all data files are uploaded as 'Supporting Information' and are invariably referred to (in the manuscript, figure legends, and the Description field when uploading your files) using the following format verbatim: S1 Data, S2 Data, etc. Multiple panels of a single or even several figures can be included as multiple sheets in one excel file that is saved using exactly the following convention: S1_Data.xlsx (using an underscore).

B) Deposition in a publicly available repository. Please also provide the accession code or a reviewer link so that we may view your data before publication. 

Regardless of the method selected, please ensure that you provide the individual numerical values that underlie the summary data displayed in the following figure panels as they are essential for readers to assess your analysis and to reproduce it: Figures 1B, 2, 3, 4, and Supplementary Figures S1, S2, S3, S4, S5, S6, S7, S8, S9, S10, S11, S12, S13, S14, S15, S16, S17, S18, S19, S20, S21, S22, S23, S24, S25AB, S26AB, S27.

3. CODE POLICY

Per journal policy, as the code that you have generated is important to support the conclusions of your manuscript, we require that you make it available without restrictions upon publication. Please ensure that the code is sufficiently well documented and reusable, and that your Data Statement in the Editorial Manager submission system accurately describes where your code can be found.

Please note that sole deposition of data or code to GitHub would not be compliant with our policies, as this could be changed after publication (https://journals.plos.org/plosbiology/s/data-availability). However, once the data/code is final, you can archive your publicly available GitHub data to Zenodo. Once you do this, it will also generate a DOI number that you can provide us with. See the process for doing this here: https://docs.github.com/en/repositories/archiving-a-github-repository/referencing-and-citingcontent

4. Please provide a blurb which (if accepted) will be included in our weekly and monthly Electronic Table of Contents, sent out to readers of PLOS Biology, and may be used to promote your article in social media. The blurb should be about 30-40 words long and is subject to editorial changes. It should, without exaggeration, entice people to read your manuscript. It should not be redundant with the title and should not contain acronyms or abbreviations.

5. We suggest a change in the title: "Combined analyses of within-host SARS-CoV-2 viral kinetics and information on past exposures to the virus in a human cohort identifies intrinsic differences of Omicron and Delta variants".

We expect to receive your revised manuscript within two weeks. 

*Published Peer Review History*

*Press*

Sincerely,

Paula

---

Senior Editor,

pjaureguionieva@plos.org,

PLOS Biology

Reviewer remarks:

Reviewer #1: All my comments have been addressed. I have no further comment

It would be worth verifying the number of patients in Table 1. Line 155, 135 participants had 1 infection episode, l140 140 participants had 1 infection episode

Reviewer #2: Very happy to recommend that this article be accepted. I look forward to seeing it published

---

## [Editor Report · Decision Letter 3]

7 Dec 2023

Dear Dr Russell,

Thank you for the submission of your revised Research Article "Combined analyses of within-host SARS-CoV-2 viral kinetics and information on past exposures to the virus in a human cohort identifies intrinsic differences of Omicron and Delta variants" for publication in PLOS Biology. On behalf of my colleagues and the Academic Editor, Roland Regoes, I am pleased to say that we can in principle accept your manuscript for publication, provided you address any remaining formatting and reporting issues. These will be detailed in an email you should receive within 2-3 business days from our colleagues in the journal operations team; no action is required from you until then. Please note that we will not be able to formally accept your manuscript and schedule it for publication until you have completed any requested changes.

Please note that my colleagues will also ask you to ensure that the figure legends in your manuscript include information on where the underlying data can be found. For example: Data underlying the graphs in this figure can be found in https://zenodo.org/records/10223386.

PRESS

Sincerely,

Christian 

Christian Schnell, PhD, 

Senior Editor

PLOS Biology

cschnell@plos.org